# Mechanisms and functions of respiration-driven gamma oscillations in the primary olfactory cortex

Joaquin Gonzalez[1,2]*, Pablo Torterolo[1], Adriano BL Tort[2]*

[1]Departamento de Fisiología, Facultad de Medicina, Universidad de la República, Montevideo, Uruguay; [2]Brain Institute, Federal University of Rio Grande do Norte, Natal, Brazil

**Abstract** Gamma oscillations are believed to underlie cognitive processes by shaping the formation of transient neuronal partnerships on a millisecond scale. These oscillations are coupled to the phase of breathing cycles in several brain areas, possibly reflecting local computations driven by sensory inputs sampled at each breath. Here, we investigated the mechanisms and functions of gamma oscillations in the piriform (olfactory) cortex of awake mice to understand their dependence on breathing and how they relate to local spiking activity. Mechanistically, we find that respiration drives gamma oscillations in the piriform cortex, which correlate with local feedback inhibition and result from recurrent connections between local excitatory and inhibitory neuronal populations. Moreover, respiration-driven gamma oscillations are triggered by the activation of mitral/tufted cells in the olfactory bulb and are abolished during ketamine/xylazine anesthesia. Functionally, we demonstrate that they locally segregate neuronal assemblies through a winner-take-all computation leading to sparse odor coding during each breathing cycle. Our results shed new light on the mechanisms of gamma oscillations, bridging computation, cognition, and physiology.

**\*For correspondence:**
joaqgonzar@gmail.com (JG);
tort@neuro.ufrn.br (ABLT)

**Competing interest:** The authors declare that no competing interests exist.

## Editor's evaluation

This fundamental study employs a publicly available dataset to examine the role of γ oscillations in the coding of olfactory information in the mouse piriform cortex. The authors convincingly show that γ originates in the piriform cortex, is driven by feedback inhibition, and that the time course of odour decoding is most accurate when γ oscillations are strongest. This work is relevant to a wide audience interested in the mechanisms and role of oscillations in the brain, and nicely demonstrates the benefits of well-curated, publicly available datasets.

## Introduction

Since the pioneer works of *Adrian, 1942* and *Bressler and Freeman, 1980*, gamma oscillations have been one of the most studied brain rhythms (*Bastos et al., 2020*; *Bastos et al., 2015*; *Bragin et al., 1995*; *Buzsáki and Wang, 2012*; *Csicsvari et al., 2003*; *Fries et al., 2007*; *Gray et al., 1989*; *Sirota et al., 2008*; *Vinck et al., 2010*; *Womelsdorf et al., 2012*; *Womelsdorf et al., 2006*). Gamma is believed to be critical for a variety of cognitive functions such as sensory processing (*Fries et al., 2001*), memory (*Fernández-Ruiz et al., 2021*), navigation (*Colgin et al., 2009*), and conscious awareness (*Rodriguez et al., 1999*). At the cellular scale, gamma rhythms modulate spiking activity, shaping the formation of transient neuronal partnerships, the so-called cell assemblies (*Buzsáki, 2010*). Computational models show that gamma depends on local inhibitory-inhibitory or excitatory-inhibitory interactions (*Tort et al., 2007*; *Wang and Rinzel, 1992*) and ultimately emerges from synchronous inhibitory

**eLife digest** The cerebral cortex is the most recently evolved region of the mammalian brain. There, millions of neurons can synchronize their activity to create brain waves, a series of electric rhythms associated with various cognitive functions. Gamma waves, for example, are thought to be linked to brain processes which require distributed networks of neurons to communicate and integrate information.

These waves were first discovered in the 1940s by researchers investigating brain areas involved in olfaction, and they are thought to be important for detecting and recognizing smells. Yet, scientists still do not understand how these waves are generated or what role they play in sensing odors.

To investigate these questions, González et al. used a battery of computational approaches to analyze a large dataset of brain activity from awake mice. This revealed that, in the cortical region dedicated to olfaction, gamma waves arose each time the animals completed a breathing cycle – that is, after they had sampled the air by breathing in. Each breath was followed by certain neurons relaying olfactory information to the cortex to activate complex cell networks; this included circuits of cells known as feedback interneurons, which can switch off weakly activated neurons, including ones that participated in activating them in the first place. The respiration-driven gamma waves derived from this 'feedback inhibition' mechanism.

Further work then examined the role of the waves in olfaction. Smell identification relies on each odor activating a unique set of cortical neurons. The analyses showed that gamma waves acted to select and amplify the best set of neurons for representing the odor sensed during a sniff, and to quieten less relevant neurons.

Loss of smell is associated with many conditions which affect the brain, such as Alzheimer's disease or COVID-19. By shedding light on the neuronal mechanisms that underpin olfaction, the work by González et al. could help to better understand how these impairments emerge, and how the brain processes other types of complex information.

postsynaptic potentials (*Buzsáki and Wang, 2012*). However, despite the initial evidence for their underlying principles being general, gamma oscillations are not a monolithic entity but actually encompass a diversity of rhythms observed experimentally (*Lopes-Dos-Santos et al., 2018*; *Scheffer-Teixeira et al., 2012*; *Schomburg et al., 2014*; *Zhong et al., 2017*). This warrants studying the mechanisms and functions of these oscillations in specific brain regions in vivo.

Gamma oscillations usually appear nested within slower rhythms, a phenomenon known as cross-frequency coupling, in which the amplitude of gamma waxes and wanes depending on the phase of a slow oscillation (*Canolty and Knight, 2010*; *Lisman and Jensen, 2013*). Important examples are the coupling of specific gamma sub-bands to the hippocampal theta rhythm (*Cavelli et al., 2020*; *Sirota et al., 2008*; *Tort et al., 2009*; *Tort et al., 2008*) or to the phase of breathing cycles (*Cavelli et al., 2018*; *Ito et al., 2014*; *Zhong et al., 2017*). Regarding the latter, it is worth noting that respiration-entrained brain rhythms depend on nasal airflow and are not a consequence of the respiratory pattern generation in the brainstem (*Lockmann et al., 2016*; *Moberly et al., 2018*; *Yanovsky et al., 2014*). Therefore, it seems likely that respiration-entrained gamma activity arises from local computations driven by sensory inputs sampled at each breath and thus plays a major role in cognition.

A promising area to study this hypothesis is the piriform cortex (PCx), which constitutes the primary olfactory area in the rodent brain (*Bolding and Franks, 2018a*; *Stettler and Axel, 2009*) and exhibits prominent gamma oscillations (*Bressler and Freeman, 1980*; *Courtiol et al., 2019*; *Freeman, 1960*; *Freeman and Skarda, 1985*; *Kay et al., 2009*; *Kay and Freeman, 1998*; *Litaudon et al., 2008*; *Mori et al., 2013*; *Vanderwolf, 2000*). Most of our understanding of piriform oscillations comes from the studies of Walter Freeman in the 20th century (*Barrie et al., 1996*; *Eeckman and Freeman, 1990*; *Freeman, 1968*; *Freeman, 1960*; *Freeman, 1959*). Freeman characterized gamma activity in terms of its topography, frequency range, and relationship to unitary activity and behavior. These observations led him to hypothesize that these oscillations constitute a fundamental sensory processing component that emerges from an excitatory-inhibitory feedback loop. However, despite his influential insights, Freeman's conjectures could not be conclusively tested due to the technological limitations of his time. Thus, we still lack compelling experimental demonstrations

of how gamma generation depends on the interactions between the different piriform neuronal subpopulations or how gamma relates to odor representations encoded as cell assemblies. Therefore, the mechanisms of gamma oscillations in the PCx and their functional role in olfaction need to be studied under the lens of modern experimental and analytical tools (*Courtiol et al., 2019*; *Kay et al., 2009*; *Mori et al., 2013*).

In this report, we study gamma oscillations in the PCx of the awake mouse. We took advantage of modern genetic tools, which accurately identify neuronal populations and precisely modify the local connectivity of the PCx, thus enabling an unprecedented study of the mechanisms and functions of its oscillatory activity. We found that respiration drives gamma oscillations in this region, which derive from feedback inhibition and depend on recurrent connections between local excitatory and feedback inhibitory populations. This loop is triggered by the projection of mitral/tufted cells in the olfactory bulb onto the principal cells of the PCx. As functional consequences, we show that respiration-driven gamma oscillations determine odor-assembly representations through a winner-take-all computation taking place within breathing cycles.

## Results
### Respiration drives gamma oscillations in the piriform cortex

To understand the mechanisms and functions of gamma oscillations and their relationship with respiration in the mouse brain, we analyzed local field potentials (LFP) from the PCx recorded simultaneously with the respiration signal (*Figure 1A*) during odorless cycles, hereafter referred to as 'spontaneous' activity. The dataset was collected by *Bolding and Franks, 2018a* and generously made available through CRCNS (http://crcns.org, pcx-1 dataset). *Figure 1B* depicts the LFP and respiration signal from a representative animal; notice that low-gamma oscillations (30–60 Hz) emerge following inhalation start. These low-gamma oscillations, already evident in raw recordings, are part of a larger gamma peak (30–100 Hz) in the LFP power spectrum (*Figure 1C*), reflecting a true oscillation (*Yuval-Greenberg et al., 2008*). Consistent with *Figure 1B*, only the low-gamma sub-band couples to the respiration cycle across animals, as its amplitude is modulated by both a 2–3 Hz LFP rhythm coherent to respiration (*Figure 1D* top panel; see also *Figure 1—figure supplements 1–3*) and by respiration itself (*Figure 1D* bottom panel). Thus, respiration entrains low-gamma oscillations in the PCx.

To further characterize the interaction between respiration and low-gamma oscillations in the PCx, we performed directionality analyses (*Figure 1E* and *Figure 1—figure supplement 4*). The gamma amplitude envelope showed a peak ~200 ms following the inhalation start, coinciding with a large positive cross-correlation peak between these signals, which suggests that respiration causes gamma. Consistent with these results, time-domain Granger causality was significantly higher in the respiration→gamma direction than in the opposite one ($t(12) = 9.82$, $p < 10^{-6}$). Together, these results show that respiration drives low-gamma oscillations in the PCx.

We analyzed the contribution of the different PCx neuronal populations to network low-gamma oscillations. First, we performed a current source density analysis (*Figure 1F*), which revealed that these oscillations are generated locally within the piriform circuit and show a phase reversal near layer 2 (where pyramidal cells are located). Next, we classified single units according to the expression of a light-sensitive channelrhodopsin coupled to the vesicular GABA transporter (VGAT). This allowed us to discriminate between VGAT- principal cells and VGAT+ inhibitory interneurons. Additionally, VGAT+ neurons were further classified into feedback inhibitory interneurons (FBI) and feedforward interneurons (FFI) according to their location relative to the principal cell layer (FFI are located at layer 1 while FBI tend to be located within the cell layer 2/3; *Bolding and Franks, 2018a*; *Figure 1—figure supplement 5*). Upon averaging the activity of each neuronal subpopulation, we found that the time course of FBI firing rate changes correlates with the amplitude of low-gamma oscillations in time (*Figure 1G*). In contrast, principal cells and FFI spike earlier within the respiratory cycle and return to baseline during the gamma amplitude peak (*Figure 1G*). Notice further that the gamma peak coincides with principal cell inhibition, as evidenced by their firing rate decrease. Moreover, by computing spike-triggered gamma envelope averages, we confirmed that gamma oscillations closely follow FBI spiking (*Figure 1G* inset). Therefore, we conclude that respiration-driven low-gamma oscillations in the PCx arise from feedback inhibition.

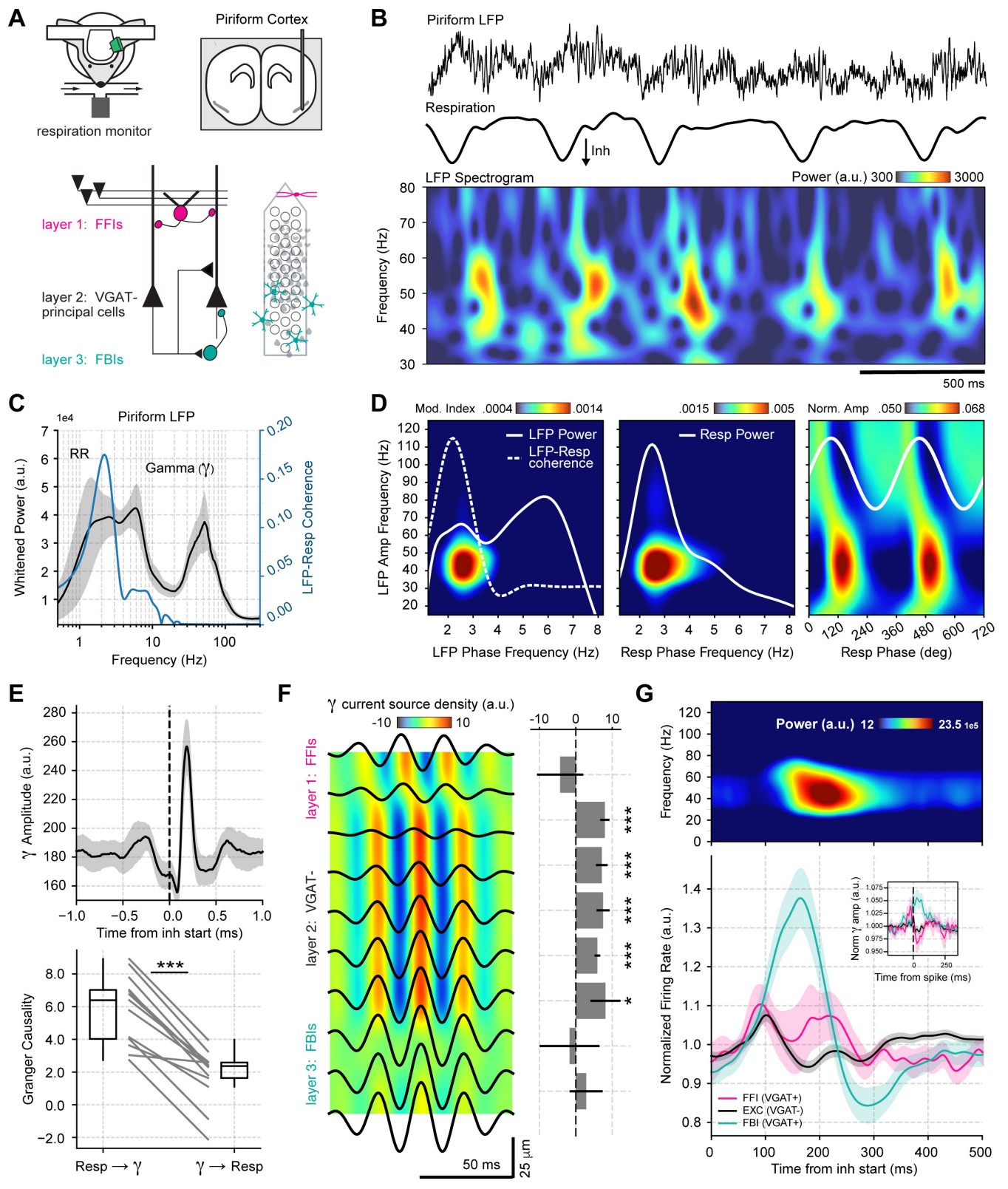

**Figure 1.** Respiration drives feedback inhibition-based gamma oscillations in the piriform cortex. (**A**) Experimental scheme, probe localization, and diagram of the local piriform circuit (modified from *Bolding and Franks, 2018a*). (**B**) Example of simultaneously recorded local field potentials (LFP) (top) and respiration (middle) signals, along with the LFP wavelet spectrogram (bottom). Notice prominent rhythmical appearance of gamma oscillations. (**C**) Average LFP power spectrum (± 2*SEM; n=13 recording sessions from 12 mice). The spectrum was whitened by multiplying each

*Figure 1 continued*

value by the associated frequency. Average LFP-Respiration coherence is superimposed in blue. (**D**) Average phase-amplitude comodulogram using either the LFP (left) or the respiration (Resp; middle) phase. Superimposed white lines show the LFP or Resp power spectrum (solid) and the LFP-Resp coherence (dashed). The right panel shows the normalized amplitude for LFP-filtered frequency components as a function of the Resp phase (average over n=13 recording sessions from 12 mice). (**E**) Directionality analyses between Resp and the gamma envelope (30–60 Hz). Shown are the average (± SEM, n=13 recording sessions from 12 mice) gamma envelope triggered by inhalation start (top), and the Granger causality for the Resp→gamma and gamma→Resp directions (bottom; boxplots show the median, 1st, 3rd quartiles, and the distribution range; each dot shows an individual mouse). (**F**) Average current source density for the gamma band (n=13 recording sessions from 12 mice). Superimposed black lines show the average gamma waveforms for each recording site. Bar plots depict statistical comparisons against a zero-current distribution (mean ± SEM; n=13 recording sessions from 12 mice). (**G**) Respiration-evoked LFP responses. Top: average inhalation-triggered whitened spectrogram (n=15 recording sessions from nine mice). Bottom: Normalized spike rate (mean ± SEM) of excitatory (EXC; VGAT-, 858 neurons), feedback inhibitory (FBI; VGAT+, 40 neurons), and feedforward inhibitory (FFI; VGAT+, 13 neurons) neuronal populations triggered by inhalation. Inset shows the normalized spike-triggered gamma amplitude envelope for each neuronal subpopulation (mean ± SEM). Normalization consisted of dividing the triggered gamma amplitude values by the mean amplitude 500 ms before each spike.

The online version of this article includes the following figure supplement(s) for figure 1:

**Figure supplement 1.** Similar respiratory frequency ranges across mice.

**Figure supplement 2.** The 1–3 Hz local field potentials (LFP) band is entrained by respiration.

**Figure supplement 3.** Respiration-driven gamma oscillations in the piriform cortex are evident across recording sites.

**Figure supplement 4.** Respiration leads low-gamma oscillations in the piriform cortex.

**Figure supplement 5.** Position and average waveform of neuronal subtypes.

## Respiration-driven gamma oscillations depend on recurrent connections within the piriform cortex

We next studied the circuit mechanisms responsible for the respiration-driven gamma oscillations in the PCx. To that end, we analyzed spontaneous PCx LFPs following the selective expression of the tetanus toxin light chain in principal cells of a targeted hemisphere (TeLC ipsi; *Figure 2A*). Under this approach, TeLC expression blocks excitatory synaptic transmission without affecting cellular

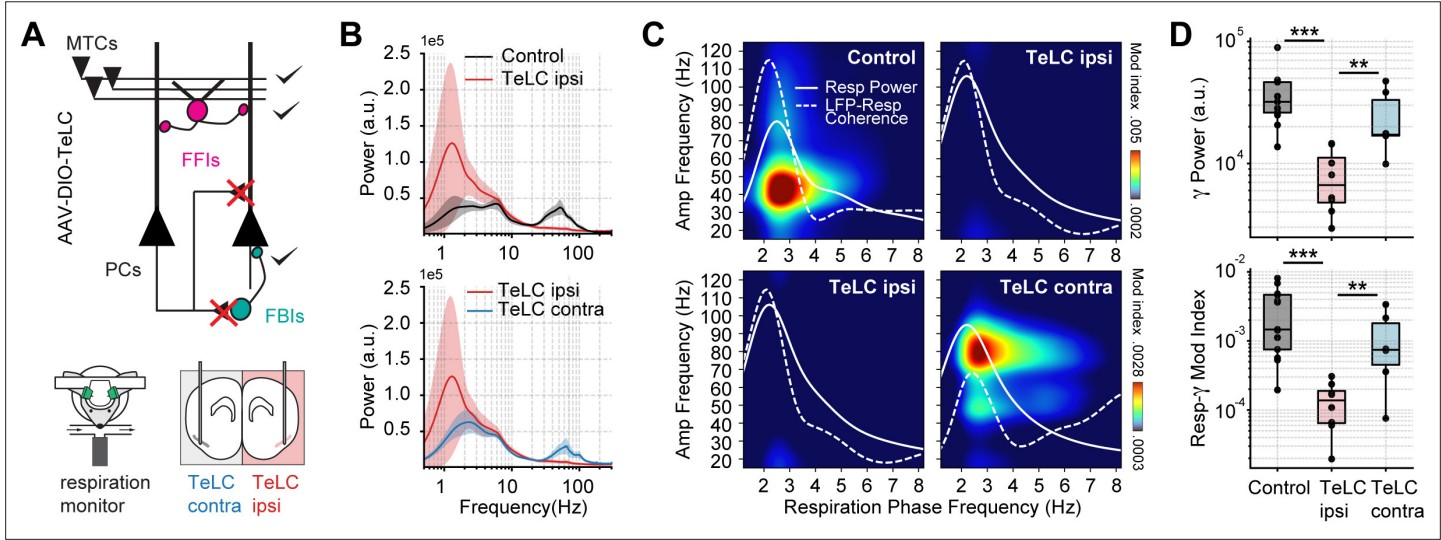

**Figure 2.** Respiration-driven gamma oscillations depend on recurrent connections within the piriform cortex. (**A**) Schematic of circuit changes after TeLC expression in principal cells (PCs) of the piriform circuit (MTCs: mitral cells; FFIs: feedforward interneurons; FBIs: feedback interneurons). Recordings were made both ipsi- and contralaterally to the TeLC expression (modified from ***Bolding and Franks, 2018a***). (**B**) Average (± 2*SEM) power spectra for control and TeLC-infected animals (Control, n=13 recording sessions from 12 mice; TeLC ipsi, n=8 recording sessions from eight mice; TeLC contra, n=6 recording sessions from 6 mice). Notice that local TeLC expression abolishes ipsilateral gamma oscillations in the PCx. (**C**) Average respiration-LFP comodulograms for control and TeLC-infected animals. Respiration power and LFP-respiration coherence are shown superimposed (same scale across plots). (**D**) Boxplots showing gamma power (top) and the Resp-low gamma modulation index (bottom) for control and TeLC-infected animals.

The online version of this article includes the following figure supplement(s) for figure 2:

**Figure supplement 1.** Ketamine/xylazine anesthesia abolishes spontaneous low-gamma oscillations in the piriform cortex.

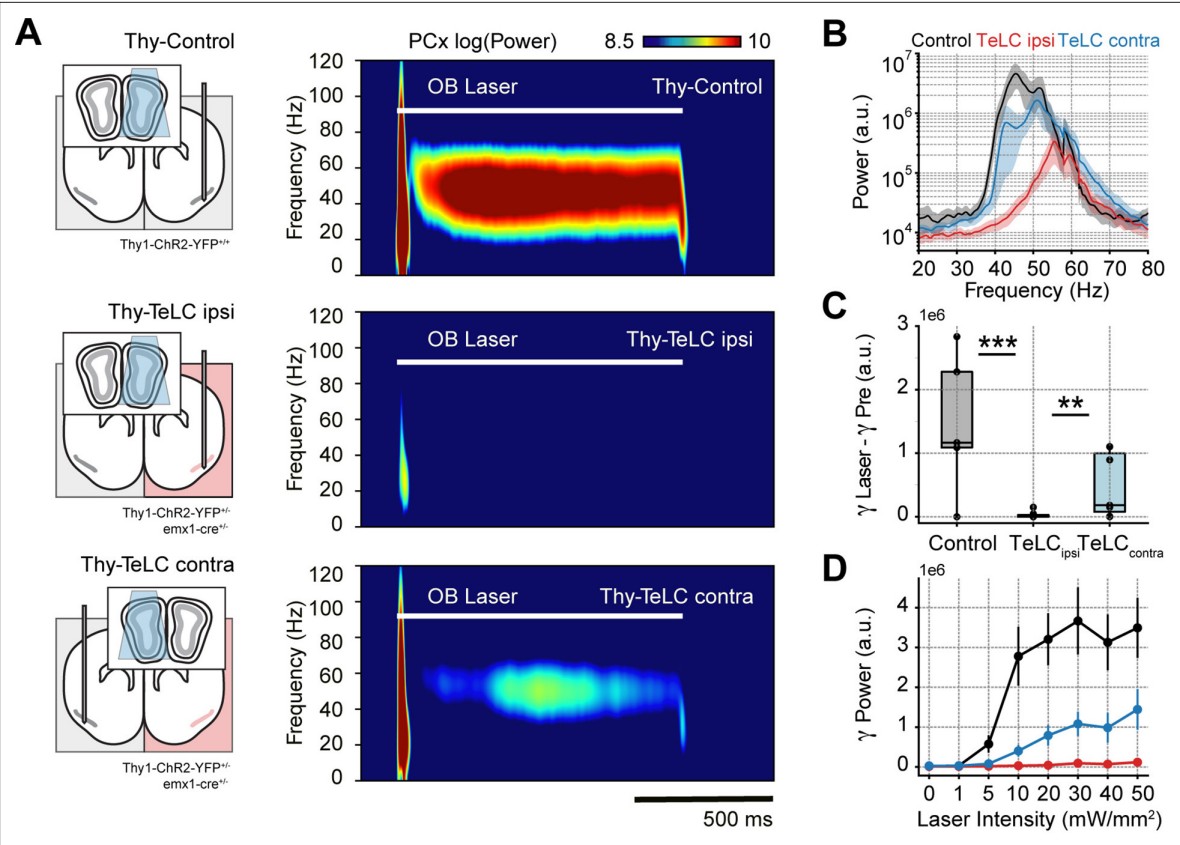

**Figure 3.** Piriform recurrent connections are necessary for olfactory bulb (OB) mitral/tufted cells to trigger low-gamma oscillations. (**A**) Left: experimental conditions for each group. Right: Average piriform cortex (PCx) spectrograms during optogenetic stimulation of the olfactory bulb (OB). (**B**) Average gamma power during OB stimulation for the control (n=5 recording sessions from five mice), TeLC ipsi (n=14 recording sessions from eight mice) and contralateral recordings (n=7 recording sessions from five mice). Note that a logarithmic y-axis is employed here while subsequent plots use a linear scale. (**C**) Boxplots showing the gamma power difference between the laser and pre-laser periods. (**D**) Gamma power as a function of the laser intensity for each experimental condition (mean ± SEM).

excitability (**Bolding and Franks, 2018a**), allowing us to study the local computations underlying the respiration-driven gamma. **Figure 2B** shows that TeLC ipsi LFPs had a large significant reduction in low-gamma oscillations with respect to either control mice (t(19) = 4.11, p<0.001) or the contralateral PCx (TeLC contra, not infected; t(12) = 2.99, p=0.0055). Moreover, TeLC ipsi LFPs also showed a significant decrease in respiration-gamma coupling compared to control (t(19) = 5.87, p<$10^{-5}$) or TeLC contra LFPs (t(12) = 3.10, p=0.0045) (**Figure 2C and D**), despite the respiration signal still reaching the PCx (note the dotted white traces in **Figure 2C** showing LFP-Resp coherence). These results demonstrate that gamma oscillations and their coupling to respiration depend on local recurrent excitatory connections within the PCx. Noteworthy, respiration-driven low-gamma oscillations also depend on the cognitive state since ketamine/xylazine anesthesia abolishes them (**Figure 2—figure supplement 1**, Appendix 1).

## Olfactory bulb mitral-cell projections trigger feedback inhibition-based gamma oscillations in the piriform cortex

After confirming that respiration-driven gamma oscillations depend on recurrent connections formed by principal cells, we asked how PCx inputs affect gamma generation. We expected mitral/tufted cell activation in the olfactory bulb (OB) to trigger similar low-gamma oscillations since these projections convey the respiratory inputs to the PCx (**Pashkovski et al., 2020**). Consistently, optogenetic activation of the OB (Thy-Control) triggered piriform low-gamma oscillations, which matched the laser time course (**Figure 3A**, top panel). Interestingly, TeLC ipsi LFPs showed almost no gamma activity following laser onset (**Figure 3A**, middle panel), while TeLC contra LFPs still exhibited gamma activity

upon stimulation of the OB (*Figure 3A*, bottom panel). Analyzing the group response, we found a significant reduction of low-gamma activity following the laser onset in TeLC ipsi LFPs compared to control (t(17) = 5.12, p<0.0001) or TeLC contra LFPs (t(19) = 3.38, p=0.0015) (*Figure 3B and C*), further confirming the importance of recurrent connections for gamma generation. Moreover, low-gamma power in control and TeLC contra LFPs increased with laser intensity, while it remained constant in TeLC ipsi LFPs (*Figure 3D*). Nevertheless, it should be noted that the contralateral TeLC hemisphere showed lower amplitude gamma oscillations following light stimulation than control recordings, though whether this gamma difference is related to an impaired network interplay between both hemispheres or to genetic differences between mouse lines remains to be determined. In any event, these results experimentally prove two critical facts about respiration-driven piriform gamma oscillations. First, that respiration drives PCx low-gamma oscillations mediated by OB projections from mitral/tufted cells. Second, feedforward interneurons do not generate low gamma, which necessarily requires the principal cells to excite local feedback interneurons.

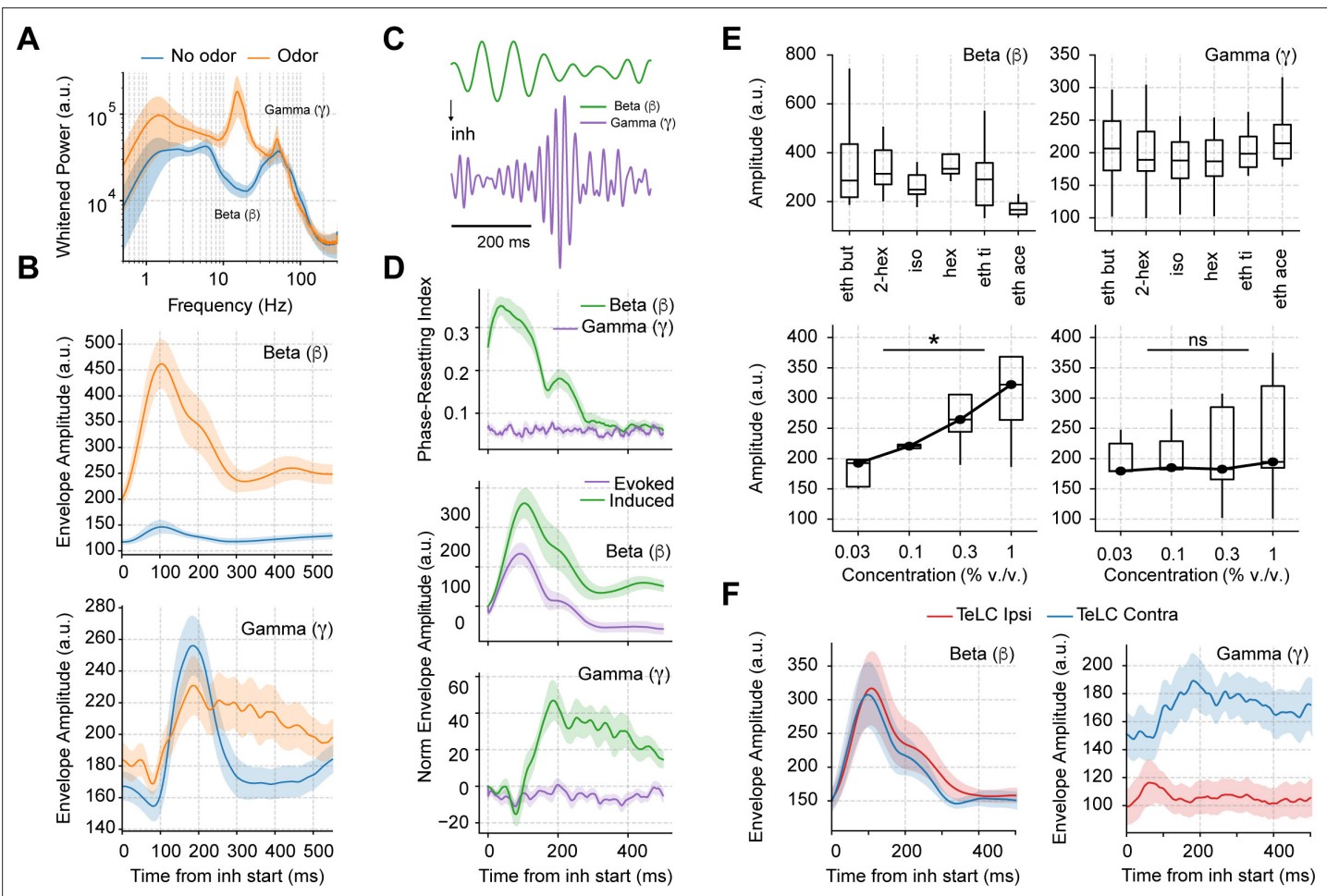

**Figure 4.** Odor delivery evokes beta and induces longer lasting gamma oscillations. (**A**) Average whitened local field potentials (LFP) power spectrum for odor and odorless respiration cycles (± 2*SEM; n=13 recording sessions from 12 mice). (**B**) Top: Average beta (top) and gamma (bottom) amplitude for odorless respiratory cycles (blue) and for cycles with odor delivery (orange). (**C**) Filtered beta (10–20 Hz) and gamma (30–60 Hz) oscillations during odor delivery. (**D**) Top: Phase-resetting index for each oscillation. Middle: Normalized induced (green) and evoked (purple) beta (middle) and gamma (bottom) amplitude triggered by inhalation. The normalization consisted of removing the average amplitude across time. All results obtained during odor delivery. Traces show mean ± SEM. (**E**) Average beta (left) and gamma (right) amplitude during odor cycles. Top panels show the average amplitude for different odorants at the same concentration (0.3% v./v., n=13 recording sessions from 12 mice). Bottom panels show the response to increasing odor concentrations (amplitudes averaged for ethyl butyrate and hexanal odorants; n=5 recording sessions from fivemice). (**F**) Amplitude envelopes during odor cycles for beta (left) and gamma oscillations (right) in TeLC experiments. Shades represent the mean ± SEM.

The online version of this article includes the following figure supplement(s) for figure 4:

**Figure supplement 1.** Longer breaths do not account for prolonged gamma activity in response to odors.

## Odors induce long-lived gamma oscillations

Having studied the mechanisms of spontaneous respiration-driven low-gamma oscillations in awake mice, we next analyzed their behavior during odor sampling (*Figure 4*). First, we noted that odors elicited large beta oscillations (10–30 Hz), which have been widely associated with olfactory processing (*Lepousez and Lledo, 2013*; *Martin et al., 2006*; *Poo and Isaacson, 2009*). Interestingly, the amplitude of gamma oscillations was not affected by odor delivery, but the duration of gamma activity increased notoriously (*Figure 4B*). Of note, the increase in gamma duration specifically depended on odor stimulation and not on longer breaths as previously suggested by *Vanderwolf, 2000*, since longer odorless cycles did not prolong the induced gamma activity (*Figure 4—figure supplement 1*).

Notably, beta oscillations occurred 100 ms before gamma onset, and, moreover, showed a consistent phase resetting during each sniff cycle (*Figure 4C and D* top), resulting in a large amplitude envelope of the inhalation-trigged average of beta-filtered LFPs (*Figure 4D* middle), which is to say that beta was evoked at each cycle. The unfamiliar reader is referred to *Tallon-Baudry and Bertrand, 1999* for a discussion about evoked vs. induced oscillations. In short, an oscillatory activity that shows up in the average filtered trace is said to be evoked since this requires phase consistency (or 'resetting') following each stimulus. On the other hand, oscillations that increase in amplitude following each stimulus (in our case, an inhalation), but that exhibit phase jitters from trial to trial, cannot be properly detected in the averaged trace due to peak-trough cancellations across trials. Such oscillations, referred to as 'induced,' can only be detected by inspecting the average across all individual trial amplitudes (or spectrogram; see *Tallon-Baudry and Bertrand, 1999*). This is the case of piriform low-gamma oscillations (*Figure 4D* bottom) since they increase following each inhalation but exhibit time jitters in peak activity from cycle to cycle and no phase resetting (*Figure 4D* top). That is, our results show that gamma oscillations are not evoked but induced at each sniff cycle, contrasting therefore, with the evoked beta oscillations.

We also investigated how beta and gamma responses depended on odor identity and concentration (*Figure 4E*). Different odorants triggered similar beta and gamma responses (*Figure 4E* top). The amplitude of beta oscillations depended on concentration ($F_{(3,12)}=5.84$, $p=0.01$), while gamma amplitude did not though its variability increased ($F_{(3,12)}=2.89$, $p=0.079$; *Figure 4E* bottom). Importantly, when comparing TeLC-infected hemispheres with the contralateral ones during odor delivery, we found that only odor-induced gamma oscillations depended on the local piriform recurrent connections, while beta oscillations were still present in infected animals (*Figure 4F*). These results suggest that beta oscillations do not relate to local piriform computations and are likely of OB postsynaptic origin.

## Respiration-driven gamma oscillations determine odor-assembly representations through a winner-take-all computation

Next, we studied how gamma oscillations influenced piriform spiking patterns during odor processing. First, we compared the firing rate of each piriform neuronal subtype during odorless and odor cycles (*Figure 5A*). Notably, feedback interneurons showed the most pronounced changes during odor cycles, substantially increasing their firing rate and prolonging their spiking period above baseline, thus mirroring the increase in gamma duration. Nonetheless, we observed that the increased FBI spiking (~100 ms) preceded gamma amplitude increase (~200 ms; c.f. *Figures 4B and 5A*). This result may be related to the time required for different FBIs to synchronize their activity and generate gamma, as observed in computational models of gamma generation (*Wang and Buzsáki, 1996*).

We then studied how gamma oscillations influence principal cell spiking during odor delivery, pooling together all recorded principal cells and analyzing their spiking as a function of the respiration phase. We found that the respiration phase modulated principal cell spiking within breathing cycles (*Figure 5B*), though the preferred spiking phase differed across neurons. Interestingly, while a large proportion of cells were inhibited during the same respiration phase as the maximal gamma amplitude, some cells increased their spiking coincidently with the gamma peak. The respiratory phase preference was stable throughout the recording session (*Figure 5C*, top panel; $t(857) = -0.18$, $p=0.57$), and respiratory modulation of principal cell spiking did not change between odor and odorless cycles (*Figure 5C*, middle panel; $t(857) = -0.57$, $p=0.71$).

Remarkably, spike-gamma phase coupling increased during the processing of odors (*Figure 5C*, bottom panel; $t(857) = 9.49$, $p<10^{-19}$), suggesting that these prolonged oscillations play a role in

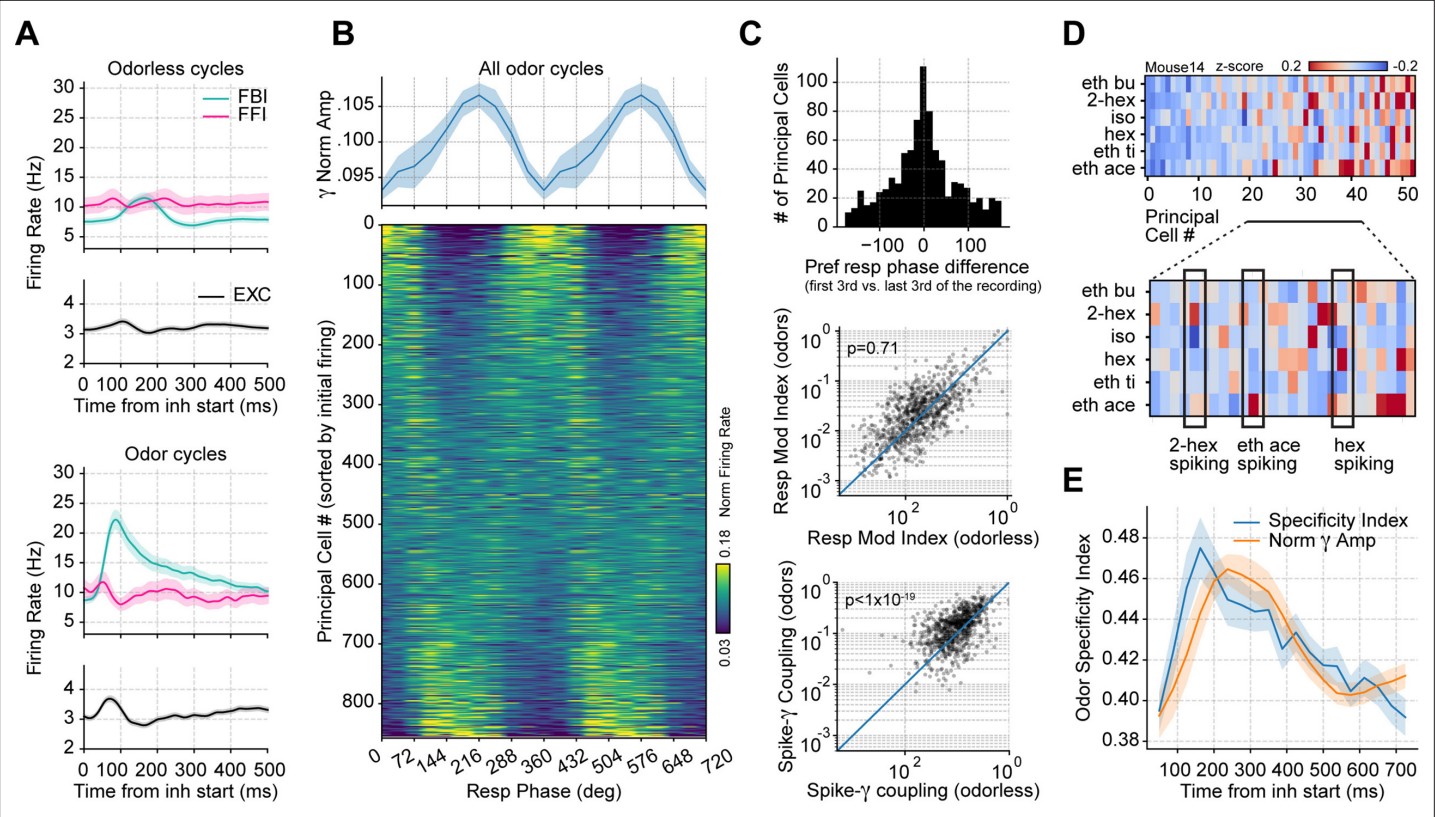

**Figure 5.** Respiration-driven gamma oscillations relate to single-cell spiking specificity to odors. (**A**) Neuronal firing rates (mean ± 0.5*SEM, n=858 EXC, 40 FBI, 13 FFI) during odorless (top) and odor cycles (bottom). (**B**) Principal cell spiking during each phase bin of the respiration cycle (bottom; 0 degree corresponds to the start of the inhalation); neurons are sorted according to the normalized firing rate in the first bin. Gamma power is shown on top (mean ± SEM, n=15 recording sessions from nine mice). (**C**) Top: preferred respiratory phase differences between the first and last thirds of the recording session (n=858 neurons). Middle: Spike-Resp coupling during odor and odorless cycles (n=858 neurons). Bottom: Spike-gamma coupling during odor and odorless cycles (n=858 neurons). (**D**) Z-scored firing rate at the gamma peak in response to different odors for a representative mouse. Columns show the firing rates of each principal cell. The bottom panel shows a zoom-in view of the differential spiking activity across odors. (**E**) Odor specificity index and normalized gamma amplitude following inhalation start (mean ± SEM, n=15 recording sessions from nine mice exposed to six different odorants at 0.3% v./v. concentration; gamma traces were rescaled to fit the plot).

shaping odor coding. Consistently, we found that odor context determined which cells fired during the gamma oscillation (*Figure 5D*). We further confirmed this observation by measuring the spiking specificity to odors, which closely followed the gamma envelope (*Figure 5E*). Thus, these results demonstrate that gamma inhibition shapes single-cell responses during olfaction.

The gamma spiking specificity supports the conjecture that respiration-driven gamma oscillations could mediate odor assembly representations. To further study this possibility, we analyzed cell assembly compositions for each odor by measuring the contribution of each principal cell to the first independent component (IC) of the population response (*El-Gaby et al., 2021*; *Lopes-dos-Santos et al., 2013*; *Trouche et al., 2016*). We found highly skewed distributions for each odor, where only a small fraction of neurons showed a strong positive contribution to the 1st IC, hereafter referred to as winner cells, while the vast majority showed low weights, referred to as loser cells (*Figure 6A and B*). Notably, the few winning neurons determining the 1st IC activity changed from odor to odor (*Figure 6A*); in other words, there was strong orthogonality in the 1st IC weight distribution across different odors. Consistent with this, the 1st IC weights were not significantly correlated between odors (corrected by multiple comparisons) (*Figure 6C*). We note that a similar odor separation was achieved when computing ICA on all odors together and analyzing the top six independent components (*Figure 6—figure supplement 1*).

Interestingly, the losing cells were significantly more phase-locked to the gamma phase (*Figure 6D* top; t(14) = 5.88, p<10$^{-4}$), consistent with them being actively inhibited during the oscillation. There

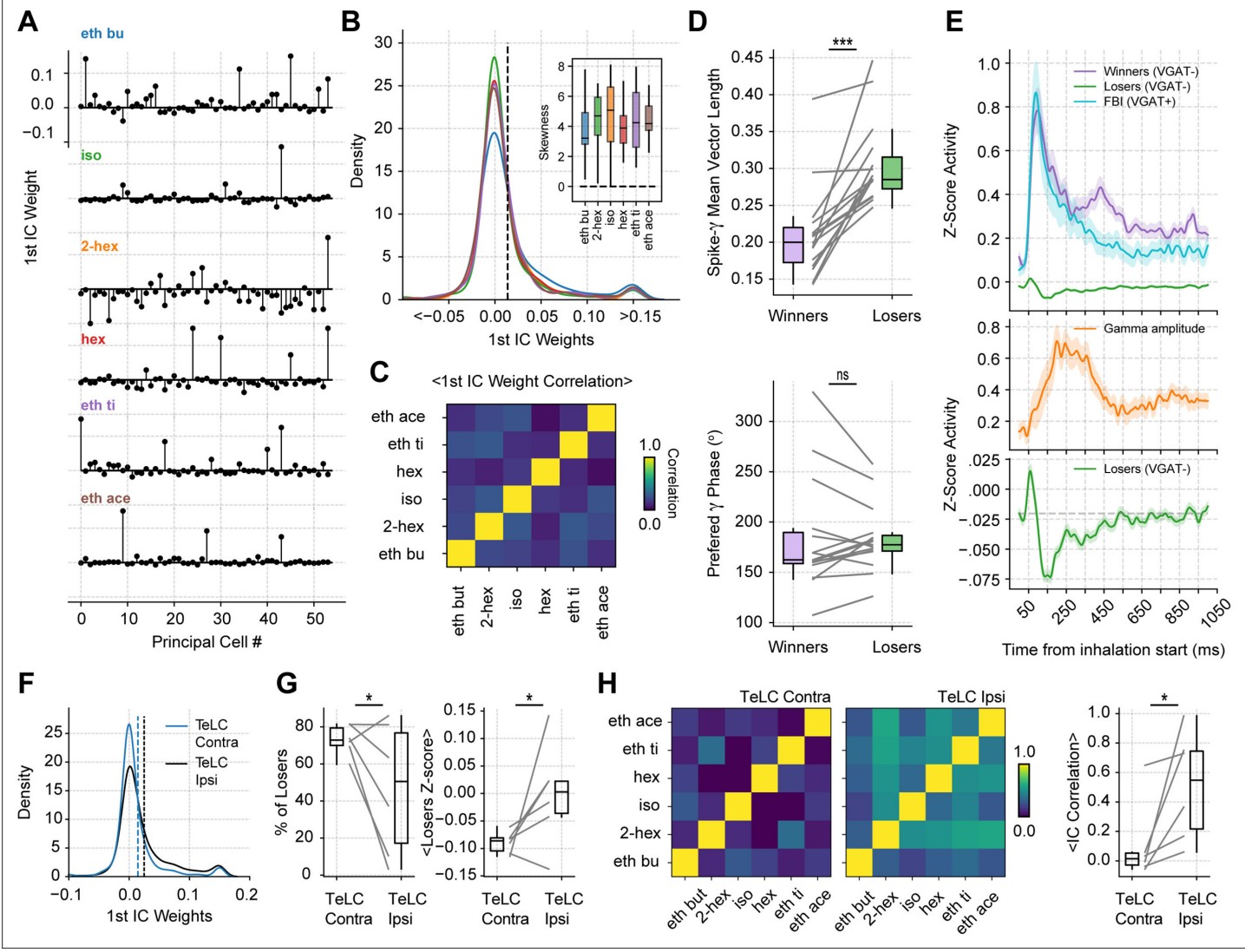

**Figure 6.** Gamma inhibition determines sparse odor-assembly representations through a winner-take-all computation. (**A**) Assembly weights for the 1st independent component (1st IC) in a representative mouse during the presentation of odorants. Notice different assembly compositions for the different odors. (**B**) Distribution of assembly weights for the 1st IC of each odor (all principal cell weights across sessions pooled together). Inset: Boxplots showing the distribution skewness for each animal and odorant. (**C**) Correlation among 1st IC weights. No pairwise odorant combination was significantly above chance (corrected for multiple comparisons). (**D**) Boxplots showing the mean-vector length of the spiking gamma phase for winning and losing neurons (top) and their preferred gamma phase (bottom). (**E**) Average z-scored spiking activity for winners, losers, feedback inhibitory neurons (top), and average z-scored gamma amplitude envelope (middle). The bottom panel shows a y-axis zoom-in view of the spiking time course of the loser neurons. n=259 winner-odor pairs, 3875 loser-odor pairs, and 40 FBI; 15 recording sessions from nine mice. (**F**) Distribution of assembly weights for the 1st IC in TeLC experiments. Each line shows the distribution average across odorants. (**G**) Percentage of losers in the infected (TeLC ipsi) and contralateral hemisphere (TeLC contra). Thresholds for defining losers were the same as employed in control recordings. (**H**) Correlation among 1st IC weights in TeLC experiments.

The online version of this article includes the following figure supplement(s) for figure 6:

**Figure supplement 1.** ICA performed on all odors simultaneously.

were no differences between the preferred gamma phase for winners and losers (*Figure 6D* bottom; t(14) = 0.05, p=0.95). Moreover, when analyzing the spiking time course of winner and loser cells, we found that both groups tended to spike immediately following inhalation start; nevertheless, once feedback inhibition was recruited, loser cells spiked below their average while the winner cells continued spiking above baseline (*Figure 6E*). Notably, loser cell inhibition correlated with gamma activity, which suggests that sparse odor assembly representations emerge from a winner-take-all process mediated by inhibitory gamma oscillations.

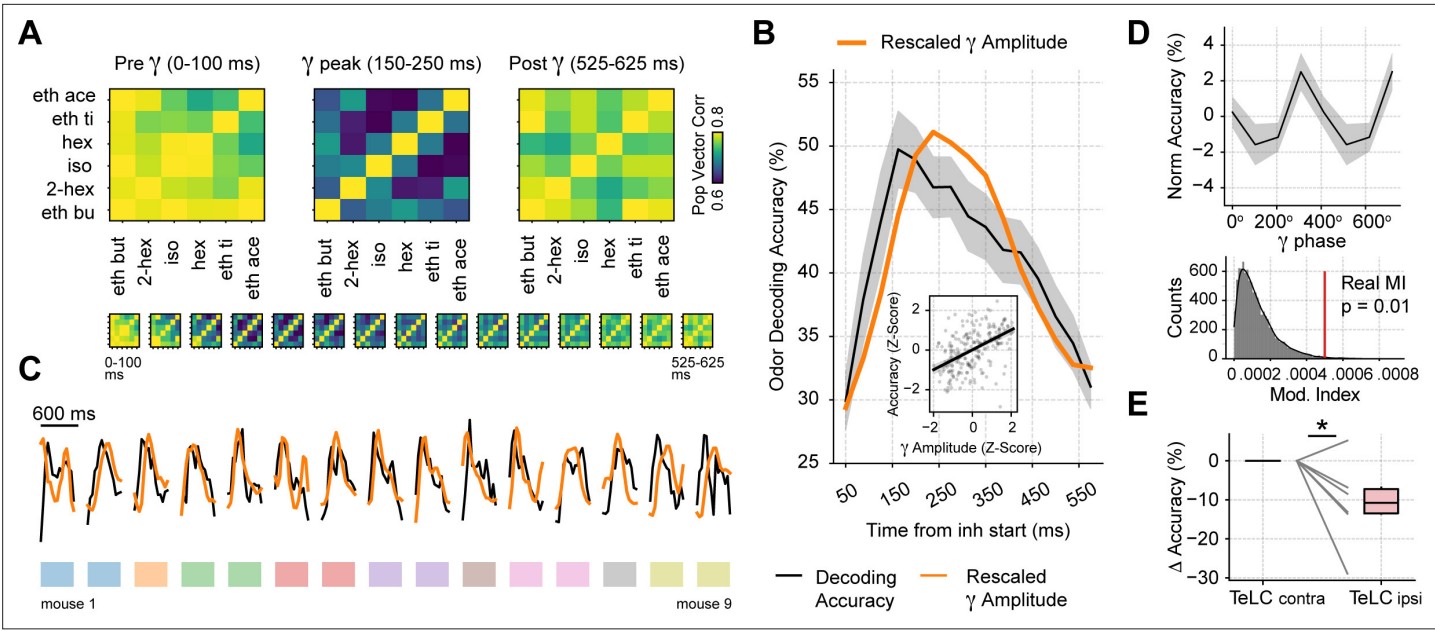

**Figure 7.** Gamma oscillations provide a privileged window for odor decoding. (**A**) Population vector correlations between odor responses during 100 ms time windows before, during and after the gamma peak (the bottom panel shows results for overlapping time windows). Only principal cells were employed for this analysis. (**B**) Odor decoding accuracy following inhalation start employing 100 ms time bins (mean ± SEM, n=15 recording sessions from nine mice). The mean accuracy across mice is shown by the black line; for comparison, the orange line shows the average gamma amplitude (arbitrary scale). The inset plot shows the correlation between gamma amplitude and odor decoding accuracy. (**C**) Odor decoding accuracy (black) and gamma amplitude (orange) time courses for each mouse. The colored rectangles underneath the traces show mouse identity. (**D**) Top: normalized odor decoding accuracy as a function of the gamma phase (mean ± SEM; n=15 recording sessions from nine mice). For each session, the normalization was obtained by subtracting the mean accuracy. Bottom: modulation index of the average decoding accuracy within the gamma cycle (Real MI) compared to a surrogate distribution (Surr MI, obtained by circularly shifting the gamma phases within each session by a random amount; n=10,000 surrogate MI values). (**E**) Odor decoding accuracy during the 150–250 times window following inhalation start in TeLC experiments (n=5 recording sessions from five mice). Boxplots show the decoding difference (delta) to the contralateral hemisphere.

The online version of this article includes the following figure supplement(s) for figure 7:

**Figure supplement 1.** Odor decoding increases during the low-gamma peak.

Because loser cell inhibition correlated with low gamma activity, we expected the winner-take-all process to be affected in TeLC recordings. Thus, we analyzed cell assemblies in both infected and contralateral hemispheres. We found that the TeLC-infected piriform cortex showed significantly less skewed distributions (t(5) = –3.08, p=0.013) with a decreased number of losing cells (t(5) = –2.10, p=0.044), which were significantly less inhibited than their contralateral counterparts (t(5) = –2.22, p=0.038; *Figure 6F and G*). Furthermore, the reduced inhibition of losing cells was associated with a high correlation of assembly weights across odors for the affected hemisphere (t(5) = 2.63, p=0.023; *Figure 6H*), suggesting that gamma oscillations are necessary for the segregation of odor-selective assemblies.

Because a winner-take-all process selects a single odor representation while actively suppressing others, it effectively implements an XOR logic gate. As a consequence, we would expect odor decoding accuracy to increase during gamma inhibition. Following this reasoning, we first analyzed the correlation of population spiking vectors in response to different odorants and found that correlations significantly dropped during the gamma peak (Gamma vs. Pre: t(14) = –3.32, p=0.002; Gamma vs. Post: t(14) = –3.73, p=0.001; *Figure 7A*), confirming that odor representations diverge the most during gamma inhibition. We next tested odor decoding from the principal cell activity by training a supervised linear classifier using population spiking (i.e. spike vectors) at different respiratory cycle phases. Consistent with a winner-take-all mechanism, we found that variations in odor decoding accuracy matched the low-gamma activity time course (*Figure 7B and C*); there was a significant correlation between these variables (r=0.45, p<10$^{-12}$). Importantly, decoding accuracy significantly increased around the gamma peak compared to time windows before or after it (Gamma vs. Pre: t(14) = 9.47,

p<10⁻⁷; Gamma vs. Post: t(14) = 7.56, p<10⁻⁵; see also *Figure 7—figure supplement 1*). This result was robust among different classification algorithms (*Figure 7—figure supplement 1*). In addition, odor decoding accuracy significantly fluctuated within the gamma cycle (*Figure 7D*), confirming that gamma inhibition directly modulates odor processing. Finally, when analyzing the TeLC experiments, we found that decoding accuracy significantly decreased in the 150–250 ms 'gamma' time window in the infected hemisphere compared to the contralateral one (t(5) = –2.41, p=0.03). In all, these findings demonstrate that respiration-driven gamma oscillations provide an optimal temporal window for olfaction.

## Discussion

The present report demonstrates the mechanisms and functions of gamma oscillations in the PCx of awake mice. We show that respiratory inputs to this cortex elicit large feedback inhibitory low-gamma oscillations that aid odor coding. Our work provides critical information for understanding gamma oscillations in the olfactory system. First, the results support the excitatory-inhibitory models of gamma generation (*Buzsáki and Wang, 2012*; *Fisahn et al., 1998*; *Freeman, 1964*), highlighting the role of excitatory neurons in initiating the gamma cycle by driving feedback interneuron spiking. Our findings thus experimentally demonstrate several of Walter Freeman's hypotheses regarding gamma generation (*Bressler and Freeman, 1980*; *Eeckman and Freeman, 1990*; *Freeman, 1968*), and reveal similarities with the mechanisms generating gamma in the OB (*Fukunaga et al., 2014*; *Fukunaga et al., 2012*; *Kay et al., 2009*; *Lepousez and Lledo, 2013*; *Mori et al., 2013*). Second, we demonstrate that the PCx low-gamma oscillations are the extracellular correlates of a winner-take-all process, thus confirming previous theoretical conjectures (*de Almeida et al., 2009*) and linking this brain oscillation to an algorithmic operation (an XOR logic gate). Third, the winner-take-all interpretation provides a direct relationship between gamma oscillations and cell assemblies.

Under this scenario, the low-gamma feedback inhibition is critical for segregating cell assemblies and generating a sparse, orthogonal odor representation. Concomitantly, the assembly recruitment would depend on OB projections determining which winner cells 'escape' gamma inhibition, highlighting the relevance of the OB-PCx interplay for olfaction (*Chae et al., 2022*; *Otazu et al., 2015*). An important feature of the respiration-driven low-gamma oscillations is that the winners triggering them change according to odor context (*Figure 6A*), a process likely dependent on odor-encoding mitral cell assemblies from the OB (*Chae et al., 2022*). Thus, the involvement of different cells and spiking dynamics would make the recruitment of feedback inhibition a non-homogenous process across respiration cycles, leading to small time jitters in peak gamma activity.

As an empirical end result, the observed low-gamma oscillations are not phase-locked from cycle to cycle; in other words, PCx low-gamma power is induced, but not evoked (*Tallon-Baudry and Bertrand, 1999*), at each respiratory cycle. Our results, nonetheless, do not exclude a role for OB gamma in triggering piriform oscillations, as the OB output may arise from similar gamma-dependent winner-take-all processes in that area. Nevertheless, our results make it less likely that piriform gamma is simply inherited from the OB, given that we observe (1) a local current source, (2) a tight correlation with FBI spiking, and (3) that gamma disappears when local recurrent connections are abolished, despite the OB remaining intact.

In contrast, beta oscillations were evoked by the respiratory input to the PCx, leading to a prominent phase resetting ~100 ms after the inhalation start (*Figure 4*). Further consistent with these results, respiration-triggered beta oscillations did not disappear in the TeLC condition and were close in time to excitatory and FFI spiking (c.f. *Figures 1, 4 and 5*). Interestingly, a previous report showed that odors evoke PCx beta oscillations under urethane anesthesia, which, as here, occurred before global inhibition (*Poo and Isaacson, 2009*). Based on this, a role for piriform beta in odor representations has been proposed; nevertheless, here we show that beta oscillations are likely inherited and do not require local computations in the PCx, i.e., they are not locally generated.

The study of the role of the PCx in olfaction has recently been boosted by selective genetic techniques, allowing researchers to examine how the individual cellular components relate to olfaction (*Franks et al., 2011*; *Nagappan and Franks, 2021*). The PCx, also known as the olfactory cortex, receives inputs from the OB and constructs a robust odor representation (*Bolding et al., 2020*). This representation can be inferred from the population activity and has two major features. On the one hand, it separates odor identity from its concentration (*Bolding and Franks, 2018a*; *Bolding and*

*Franks, 2017*; *Roland et al., 2017*). This process depends on large-scale piriform inhibition (*Bolding and Franks, 2018a*) and thus correlates with low-gamma oscillations. On the other hand, the representation groups odors into perceptual categories, not necessarily related to the physical structure of the odor molecules (*Pashkovski et al., 2020*). Surprisingly, these odor assembly representations are not fixed but change with time (*Schoonover et al., 2021*), which might be a particular property of unstructured cortices such as the piriform cortex (*Stettler and Axel, 2009*). In light of this evidence, spike-field synchronization might be an essential coding strategy, as neural oscillations might provide an internal signal in which assembly representations can be organized and structured. Thus, we hypothesize that low-gamma oscillations provide a critical time window for odor assembly representations to be formed and modified through winner-take-all computations.

The role of neuronal oscillations in shaping spiking patterns has been widely discussed (*Buzsáki, 2010*). Among other functions, oscillations are thought to provide syntactical blocks in which spiking activity can be parsed into meaningful words. A reader (receiver) area can then decode such neuronal representations by following these syntactic rules. In favor of this hypothesis, our decoding analysis shows that, for a reader neuron, optimal odor decoding occurs during the gamma peak within breathing cycles. A likely explanation for this scenario – based on our results – is that by suppressing the activity of competing representations, gamma oscillations increase the signal-to-background ratio and allow for optimal decoding. Under this framework, maximal decoding occurs after and not during the peak firing rate, coinciding with peak gamma activity and synchronous inhibition. Thus, the inhibited neurons, despite being the most phase-locked to gamma, would not directly encode odor information, while the neurons that escape gamma inhibition would do. Interestingly, and further consistent with our results, a recent study in humans showed that the accurate perception and identification of odors depend on piriform gamma oscillations (*Yang et al., 2022*). Moreover, the behavioral response to odors also seems to depend on piriform gamma, and its suppression induces depressive-like behaviors in mice (*Li et al., 2022*). Therefore, respiratory-triggered gamma oscillations might constitute a fundamental building block for neuronal communication in the olfactory system.

Several reports show that the PCx is the source of gamma activity for widespread brain regions, including the striatum (*Carmichael et al., 2017*) and the anterior limbic system (*Carmichael et al., 2019*). Although the authors largely attribute such oscillations to piriform volume conduction, recent evidence shows that olfactory gamma synchronizes distant regions as well (*Li et al., 2022*). Based on our results, we postulate that genuine respiration-entrained gamma oscillations in downstream regions are triggered by the output of the winning piriform assemblies, leading to new local gamma winner-take-all processes. This provides a new look into long-range gamma synchronization, emerging naturally if winners generating gamma in one area consistently trigger winners generating gamma in a downstream area. Such a mechanism closely agrees with the theory and experiments reported by *Schneider et al., 2021*, which suggest that oscillatory power and connectivity are the main drivers underlying inter-areal coherence.

A central element to this directional view of interregional synchronization is gamma coupling to respiration, as the slow oscillation ensures a flow of information from the most primary olfactory areas (OB, PCx) to the higher limbic areas (prefrontal cortex, amygdala). In accordance, a recent report showed that respiration-related brain oscillations drive sparse assemblies in the prefrontal cortex (*Folschweiller and Sauer, 2022*). Notably, these authors also found that inhibitory recruitment by assembly members was key for assembly segregation, suggesting a potential role for gamma in that area. Hence, the directional view of synchronization across regions places a major role in cross-frequency interactions driving communication, highlighting the relevance of interregional synchrony modulation by slower rhythms, for instance, theta (*González et al., 2020*) or respiration itself (*Cavelli et al., 2020*; *González et al., 2023*).

In light of our findings, we propose the following model to understand the relationship between spiking activity, neural oscillations, and odor coding in the piriform cortex: (1) Following each sniff, olfactory information is processed in the OB and is transmitted to the piriform circuit through mitral/tufted cell spiking. These synchronous postsynaptic potentials cause the field beta oscillation, whose amplitude depends on the afferent volley. (2) The principal cells that better encode the olfactory stimulus excite feedback interneurons. (3) Feedback interneurons inhibit competing principal cells, causing the field gamma oscillation. (4) This winner-take-all process segregates cell assemblies and dictates a sparse piriform odor representation. Note that because the piriform cortex normalized

odorant concentration (*Bolding and Franks, 2018a*), the amplitude of gamma oscillations does not change. Hence, gamma oscillations provide an optimal temporal window to decode odor.

## Materials and methods

### Datasets

We analyzed recordings of the PCx generously made available by Bolding and Franks through the Collaborative Research in Computational Neuroscience data-sharing website (http://crcns.org, pcx-1 dataset). The SIMUL, TeLC-PCx, THY, TeLC-THY, and VGAT experiments were used. Detailed descriptions of the experimental procedures can be found in previous publications (*Bolding et al., 2020*; *Bolding and Franks, 2018a*; *Bolding and Franks, 2017*). All protocols were approved by the Institutional Animal Care and Use Committee of Duke University. Below we describe the analytical methods employed by us, and, for convenience, also the relevant experimental procedures from the original publications.

### Animals

Adult mice were employed (>P60, 20–24 g), and housed in single cages on a normal light-dark cycle. For the Cre-dependent TeLC group, offspring of $Emx1^{Cre/Cre}$ breeding pairs were obtained from The Jackson Laboratory (005628). For the optogenetic experiments, the mice employed were: adult $Thy1^{ChR2/ChR2-YFP}$, line 18 (Thy1-COP4/EYFP, Jackson Laboratory, 007612) and $VGAT^{ChR2/+}$, line 8 (Slc32a1-COP4*H134R/EYFP, Jackson Laboratory, 014548). For the combined optogenetics and TeLC expression experiments, adult offspring of $Emx1^{Cre/Cre}$ mice crossed with $Thy1^{ChR2/ChR2-YFP}$ mice were employed.

### Adeno-associated viral vectors

For the TeLC experiments, AAV5-DIO-TeLC-GFP was expressed under CBA control (6/7 mice) or synapsin (1/7 mice), whose effects were similar and pooled together. Three 500 nL injections in the PCx (AP, ML, DV:+1.8, 2.7, 3.85;+0.5, 3.5, 3.8; –1.5, 3.9, 4.2; DV measured from brain surface) were employed to achieve TeLC expression. All recordings took place ~14 days after the promoter injection. All viruses were obtained from the University of North Carolina-Chapel Hill (UNC Vector Core).

### Data acquisition

The electrophysiological signals were recorded using 32-site polytrode acute probes (A1x32-Poly3-5mm- 25 s-177, Neuronexus) with an A32-OM32 adaptor (Neuronexus) through a Cereplex digital headstage (Blackrock Microsystems). For the optogenetic identification of GABAergic cells, a fiber-attached polytrode probe was employed (A1x32-Poly3-5mm-25s-177-OA32LP, Neuronexus). Data were acquired at 30 kHz, unfiltered, employing a Cerebus multichannel data acquisition system (BlackRock Microsystems). Respiration and experimental events were acquired at 2 kHz by analog inputs of the Cerebus system. The respiration signal was measured employing a microbridge mass airflow sensor (Honeywell AWM3300V), which was positioned opposite to the animal's nose. Inhalation generated a negative airflow and thus negative changes in the voltage of the sensor output.

### Electrode and optic fiber placement

A Patchstar Micromanipulator (Scientifica) was employed to position the recording probe in the anterior PCx (1.32 mm anterior and 3.8 mm lateral from bregma). Recordings were targeted 3.5–4 mm ventral from the brain surface at this position, and were further adjusted according to the LFP and spiking activity monitored online. The electrode sites spanned 275 µm along the dorsal-ventral axis. The probe was lowered until an intense spiking band was found, which covered 30–40% of electrode sites near the correct ventral coordinate, thus reflecting the piriform layer II. For the optogenetic experiments stimulating OB cells in $Thy1^{ChR2/ChR2-YFP}$ mice, the optic fiber was placed <500 µm above the OB dorsal surface.

### Spike sorting and waveform characteristics

Spyking-Circus software was employed to isolate individual units (https://github.com/spyking-circus). All clusters which had more than 1% of ISIs violating the refractory period (<2 ms) or appearing

contaminated were manually removed. Units that showed both similar waveforms and coordinated refractory periods were merged into a single cluster. The unit position was characterized as the mean electrode position (across electrodes) weighted by the amplitude of the unit waveform on each electrode.

## Odor delivery

Stimuli consisted of monomolecular odorants diluted in mineral oil. These were hexanal, ethyl butyrate, ethyl acetate, 2-hexanone, isoamyl acetate, and ethyl tiglate. Odors were presented for one second through an olfactometer controlled by MATLAB scripts and repeated every ten seconds.

## Analytical Methods

For all analyses, we used Python 3 with numpy (https://numpy.org/), scipy (https://docs.scipy.org/), matplotlib (https://matplotlib.org/), sklearn (https://scikit-learn.org/stable/), and statsmodel (https://www.statsmodels.org/stable/index.html) libraries. The codes to reproduce all seven figures are freely available at: https://github.com/joaqgonzar/Gamma_Oscillations_PCx; (copy archived at swh:1:rev:3e01d6b0112e2739fbfb6d0fef2be95f2d48ebd5) (*Gonzalez, 2023*).

## LFP Preprocessing

Raw LFPs were decimated to a sampling rate of 2000 Hz to match the respiration signal sampling rate, employing the *decimate* scipy function. This function first low-pass filters the 30 kHz raw data and then downsamples it, avoiding aliasing. For each animal, we used the same channel (either channel 17 or 16 depending on the headstage configuration) for all analyses in the SIMUL and TeLC-PCx datasets, though similar results are obtained for the rest of the channels given their high LFP redundancy (*Figure 1—figure supplement 3*). Channel 28 was employed for the VGAT datasets. We analyzed both odor-related and spontaneous activity; for the former, we selected all inhalations occurring within one-second following odor delivery, and the latter was obtained as all awake periods without odor delivery.

## Power spectrum

To analyze the LFP spectra, we computed Welch's modified periodogram using *welch* scipy function. Specifically, we employed a 1 s moving window with half a window overlap, setting the numerical frequency resolution to 0.1 Hz (by setting the nfft parameter to 10 times the sampling rate). All spectra were whitened by multiplying each power value by its associated frequency, thus eliminating the 1 /f trend. For *Figure 1B*, we computed the wavelet transform (*cwt* scipy function) of the LFP signal using a Morlet wavelet with 0.1 Hz resolution.

The triggered spectrograms were obtained first by selecting all 500 ms LFP windows following inhalation start (time stamps provided in the dataset, see *Bolding and Franks, 2018a* for further details) and then computing the spectrogram using the *spectrogram* scipy function. We used 40 ms windows with 82% overlap and a frequency resolution of 0.1 Hz. After computing each individual spectrogram, we whitened (i.e. multiplied by f) and averaged them to yield the mean spectrogram shown in *Figures 1G and 3A*.

## Phase-amplitude coupling

We computed phase-amplitude comodulograms following the methods proposed by *Tort et al., 2010*. We first band-pass filtered the LFP/respiration signal using the *eegfilt* function (*Delorme and Makeig, 2004*) adapted for Python 3 (available at https://github.com/joaqgonzar/Gamma_Oscillations_PCx (copy archived at swh:1:rev:3e01d6b0112e2739fbfb6d0fef2be95f2d48ebd5), *Gonzalez, 2023*). We filtered between 20–130 Hz in 10 Hz steps to obtain the higher frequency components, and between 1 and 8 Hz in 1 Hz steps to obtain the slower frequency components. The phase (angle) and amplitude time series of the filtered signals were estimated from their analytical representation based on the Hilbert transform. We then binned the phase time series into 18 bins and computed the mean amplitude of the fast signal for each bin. The amount of phase-amplitude coupling was estimated through the modulation index (MI) metric (*Tort et al., 2010*), MI = (Hmax-H)/Hmax, where Hmax is the maximum possible Shannon entropy for a given distribution (log(number of phase bins)) and H is the actual entropy of the amplitude distribution.

### Directionality analysis

To study the directionality between the gamma envelope and respiration signal, we employed three different strategies: (1) we averaged the gamma envelope using the inhalation start as a trigger; (2) we computed the cross-correlogram, using the correlate scipy function, between the gamma envelope and the respiration signal; (3) we computed Granger Causality estimates, using the grangercausalitytest stats model function, with a 10-order VAR model and employing the log(F-stat) as the Granger Causality magnitude (*Geweke, 1982*).

### Current-source density analysis

We first obtained gamma averages by filtering the LFP signal between 30–50 Hz; the amplitude peaks were then identified and used for averaging 100 ms epochs surrounding these timestamps. CSD analysis was obtained by -A+2B-C for adjacent probe sites.

### Induced power, evoked power, and phase-resetting index

To study induced power, we (1) filtered the recordings (beta: 10–20 Hz, low gamma: 30–60 Hz), (2) obtained their amplitude through the Hilbert transform, and (3) computed the inhalation-triggered average. To study evoked power, we (1) filtered the recordings, (2) obtained the inhalation-triggered average, and then (3) estimated the amplitude of the average signal through the Hilbert transform. For obtaining the phase-resetting index of the filtered signals, we used the same procedure as for computing the inter-trial coherence (*Makeig et al., 2004*) using 500 ms windows following inhalation start as a 'trial.'

### Individual cell spiking responses to inhalation

All cells were classified de-novo following the protocol described by *Bolding and Franks, 2018a*. A Wilcoxon rank-sum test was employed to determine laser responsiveness; $p < 0.001$ was used to detect VGAT+ neurons. The distinction between FFI and FBI was made according to their location on the probe relative to the principal cell layer (FFI are located at layer 1 while FBI tend to be located within the cell layer 2/3; *Bolding and Franks, 2018a*; *Figure 1—figure supplement 5*). The spike times were rounded to match the LFP resolution (0.5 ms or 2000 Hz sampling rate). After that, we convolved the spike times of each unit with a 10 ms standard deviation Gaussian kernel employing the *convolve* scipy function, in close similarity with the original publication, which gave rise to a smoothed spiking activity. For the normalized responses (*Figure 1G*), the spiking response of a neuronal subpopulation (i.e. excitatory, FFI, or FBI neurons) was normalized by dividing the firing rate by its average across time bins.

### Odor specificity index

We defined the odor specificity index for single cell responses following *McNaughton et al., 1983* as $SI = \max(F_x)/(F_a + F_b + F_c + F_d + F_e + F_f)$, where F represents firing rate, the subscripts a-f represent different odors, and $\max(F_x)$ is the firing rate for the odor that generates the largest response. The specificity index time course was obtained using 100 ms sliding windows (62.5% overlap) following inhalation start.

### Independent component analysis

We employed the *ICA* sklearn function on concatenated time series of the smoothed principal cell spiking activity using all 1000 ms windows following inhalation start during odor presentation (0.3% v./v. concentration). Each odor was treated separately. We analyzed only the first independent component, setting n_components = 1. Winners and losers were identified according to the weight each neuron exerted on the 1st principal component. Namely, neurons with a weight above the 95[th] percentile were labeled as winners, while neurons with a more negative weight than the IC mean as losers. For representation purposes (*Figure 6B*), large positive and negative weights were truncated at 0.15 and –0.05, respectively. The same procedures were employed for the assembly analyses on TeLC recordings, except that all cells were employed due to the lack of opto-tagging. We note that similar results were obtained in control recordings if all available neurons were included, likely due to excitatory cells far outnumbering inhibitory interneurons.

### Gamma phase-locking and preferred spiking phase

To assess the coupling between spikes and the gamma phase, the mean-vector length of the spiking gamma phases was calculated as $MVL = \|\frac{1}{N_k} \sum_{k=1}^{N_k} e^{\sqrt{-1}\varphi_k}\|$, where $\varphi_k$ is the gamma phase associated to the k-th spike, and $N_k$ is the total number of spikes. The average gamma spiking angle (stats. circmean) was computed to obtain the preferred phase. For the spike-triggered amplitude envelope averages, we first obtained the gamma amplitude envelope using the Hilbert transform, and then we computed the averages using –150–300 ms windows surrounding each spike.

### Population vector correlations

To infer the similarity/dissimilarity in the population response to the different odors, we analyzed the population vector correlations, defined as the Pearson correlation (*pearsonr* scipy function) measured using paired data from the same neurons to two odors. Correlations were computed between spiking vectors for each odor; the correlation time course shown in *Figure 7A* bottom was obtained using 100 ms sliding windows (62.5% overlap) following inhalation start. For statistics, we averaged the correlation across odors and compared them among three-time windows of interest: from 0 to 100 ms (Pre), from 150 to 250 ms (Gamma), and from 525 to 625 ms (Post).

### Odor decoding

We employed a supervised linear classifier to decode the odor identity (0.3% v./v. concentration) from the population spiking vectors. The classification algorithm was supplied with the average spiking of each principal cell in 100 ms sliding windows following the inhalation start. The classification algorithm was a linear support vector machine with a stochastic gradient descent optimization, implemented using the *sgdclassifier* sklearn function. Each mouse was trained and tested separately; for each time window, the training sample consisted of 2/3 of the data, and the test sample consisted of the remaining third. The division of training/test samples was randomized in order to avoid odor bias, repeated 100 times, and the results of the repetitions averaged to obtain the final decoding value. For the statistical comparisons, we used the same 100 ms time windows employed in the population vector correlations (i.e. Pre, Gamma, and Post). To verify the robustness of the results, we also employed a linear discriminant analysis and a k-nearest neighbor classifier, which rendered similar results to the sgdclassifier algorithm (*Figure 7—figure supplement 1*).

### Statistics

We show data as either mean ± SEM or regular boxplots showing the median, 1st, 3rd quartiles, and the distribution range without outliers. We employed paired and unpaired t-test to compare between groups. To compare among odor concentrations, we employed repeated measures ANOVA. We set $p < 0.05$ to be considered significant (in the figure panels, *, ** and *** denote $p < 0.05$, $p < 0.01$, and $p < 0.001$, respectively). For the 1st IC weight correlations, we employed Pearson r correlation and corrected the statistical threshold through the Bonferroni method for multiple comparisons.

### Data availability

All the data employed is freely available at http://dx.doi.org/10.6080/K00C4SZB. Previously Published Datasets: *Bolding and Franks, 2018b* Collaborative Research in Computational Neuroscience (http://crcns.org/). 'Simultaneous extracellular recordings from mice OB and PCx and respiration data in response to odor stimuli and optogenetic stimulation of OB (https://crcns.org/data-sets/pcx/pcx-1).'

## Acknowledgements

We thank Kevin Bolding and Kevin Franks for making their data available and for their insightful comments on our manuscript. We also thank Diego Laplagne for comments on the manuscript. JG was supported by Comision Academica de Posgrado (CAP), Programa de Desarrollo de Ciencias Básicas (PEDECIBA), and Comisión Sectorial de Investigación Científica (CSIC). PT was supported by PEDECIBA and CSIC. ABLT was supported by Conselho Nacional de Desenvolvimento Científico e Tecnológico (CNPq) and Coordenação de Aperfeiçoamento de Pessoal de Nível Superior (CAPES), Brazil.

## Additional information

### Funding

| Funder | Grant reference number | Author |
| --- | --- | --- |
| Conselho Nacional de Desenvolvimento Científico e Tecnológico | | Adriano BL Tort |
| Coordenação de Aperfeiçoamento de Pessoal de Nível Superior | | Adriano BL Tort |
| Comisión Sectorial de Investigación Científica | I+D grupos 2022- group ID-883465 | Joaquin Gonzalez Pablo Torterolo |

The funders had no role in study design, data collection and interpretation, or the decision to submit the work for publication.

### Author contributions

Joaquin Gonzalez, Conceptualization, Software, Formal analysis, Visualization, Methodology, Writing – original draft, Writing – review and editing; Pablo Torterolo, Supervision, Funding acquisition, Writing – review and editing; Adriano BL Tort, Supervision, Funding acquisition, Project administration, Writing – review and editing

### Author ORCIDs

Joaquin Gonzalez http://orcid.org/0000-0002-8721-4292
Adriano BL Tort http://orcid.org/0000-0002-9877-7816

### Ethics

The present study used a third-party dataset and required no ethical permit for the performed computational analyses. The experimental protocols of the original data source (Bolding and Franks, 2018) were approved by Duke University Institutional Animal Care and Use Committee (protocol A220-15-08).

### Decision letter and Author response

Decision letter https://doi.org/10.7554/eLife.83044.sa1
Author response https://doi.org/10.7554/eLife.83044.sa2

## Additional files

### Supplementary files

• MDAR checklist

### Data availability

All the data employed is freely available at: http://crcns.org, pcx-1 dataset: https://doi.org/10.6080/K00C4SZB.

The following previously published dataset was used:

| Author(s) | Year | Dataset title | Dataset URL | Database and Identifier |
| --- | --- | --- | --- | --- |
| Franks K, Bolding K | 2018 | Simultaneous extracellular recordings from mice olfactory bulb (OB) and piriform cortex (PCx) and respiration data in response to odor stimuli and optogenetic stimulation of OB | https://crcns.org/data-sets/pcx/pcx-1 | CRCNS.org, pcx-1 |

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

## Appendix 1

### Respiration-driven low-gamma oscillations fade under anesthesia

After unveiling the mechanisms responsible for the respiration-driven low-gamma activity in the PCx, we investigated whether these oscillations are state-dependent. To that end, we analyzed PCx LFP recordings during ketamine/xylazine anesthesia. *Figure 2—figure supplement 1* shows the LFP spectrogram before and after ketamine/xylazine administration. Anesthesia promoted large-amplitude, slow LFP oscillations while greatly reducing gamma power, an effect that lasted ~30 min (*Figure 2—figure supplement 1A*). These results can also be readily seen in the average power spectra of awake vs. anesthetized animals (*Figure 2—figure supplement 1B*), thus confirming that PCx low-gamma oscillations depend on the brain state. Next, we investigated if ketamine/xylazine anesthesia alters gamma coupling to respiration by analyzing respiration-LFP phase-amplitude comodulograms. We found that low-gamma oscillations are no longer coupled to respiration during general anesthesia (*Figure 2—figure supplement 1C*), despite the respiration-entrained rhythm being still present in the PCx LFP (*Figure 2—figure supplement 1D and E*). Intriguingly, the respiration-LFP comodulogram revealed coupling to a faster gamma activity (>80 Hz) under anesthesia (*Figure 2—figure supplement 1C*), which we deem likely to relate to the high-frequency oscillations evoked by sub-anesthetic doses of ketamine (*Caixeta et al., 2013*; *Castro-Zaballa et al., 2018*; *Hunt et al., 2019*), though it exhibited no power spectrum peak (*Figure 2—figure supplement 1B*).

### Methods

#### Ketamine/xylazine anesthesia

Ketamine/xylazine (100/10 mg/kg) was administered intraperitoneally in 11 of the 13 mice from the control dataset, inducing stable anesthesia during a 30–45 min time period. A heating pad was used to maintain body temperature during this time. For all analyses, we employed the first 30 min following the slow-oscillation onset. A single animal was discarded since no electrophysiological markers of anesthesia were observed.

#### Independent component analysis on all odors

We employed the *ICA* sklearn function on concatenated time series of smoothed principal cell spiking activity using all 100–400 ms windows following inhalation start during odor presentation for all six odors (0.3% v./v.). We analyzed the first six independent components, setting n_components = 6. Of note, the choice of the time window was motivated by inspecting the time period of peak gamma activity following inhalation. For comparison, we also calculated single-odor ICA decompositions using the same time window.

