## [Editor Report]

This fundamental study employs a publicly available dataset to examine the role of γ oscillations in the coding of olfactory information in the mouse piriform cortex. The authors convincingly show that γ originates in the piriform cortex, is driven by feedback inhibition, and that the time course of odour decoding is most accurate when γ oscillations are strongest. This work is relevant to a wide audience interested in the mechanisms and role of oscillations in the brain, and nicely demonstrates the benefits of well-curated, publicly available datasets.

---

## [Decision Letter]

**Decision letter after peer review:**

Thank you for submitting your article "Mechanisms and functions of respiration-driven γ oscillations in the primary olfactory cortex" for consideration by *eLife*. Your article has been reviewed by 2 peer reviewers, and the evaluation has been overseen by a Reviewing Editor and Laura Colgin as the Senior Editor. The reviewers have opted to remain anonymous.

Essential revisions:

All points raised by the reviewers are for clarification and should be addressable either with textual adjustments or minimal additional analysis.

While I would ask you to address all comments with textual clarifications, please do in particular consider the first two points made by reviewer 1 (directly γ-cycle dependent decoding – if feasible; decoding in the TeLC recordings, as similarly raised by reviewer 2) and the first and fifth comment of reviewer 2 (different populations and the TeLC comment as for Reviewer 1).

*Reviewer #1 (Recommendations for the authors):*

– Does the observation that winner cells fire before the peak of the γ oscillation imply that γ is unimportant for neural coding and it is the deviation from the γ cycle that matters most?

– The authors should be more explicit in the discussion of the initial results (Figures 1 and 2) that they are analyzing activity only during "spontaneous activity", i.e., recorded in the absence of odor. It took me a while to figure this out.

– Light-evoked γ in contralateral hemispheres is much lower than in contralateral control hemispheres but never discuss this. It warrants mention.

– Line 125: Confusing, FFIs in L. 1 are deeper but still more superficial than FBIs.

– There is a small but noticeable β peak coinciding w/ peak of FFI firing (Figure 1F). This is worth mentioning.

– Does it matter that the authors selectively examined activity in VGAT- neurons, as opposed to the unsorted population? Would their observations hold (e.g. winner vs. loser cells) if they did not have this sorting?

*Reviewer #2 (Recommendations for the authors):*

1. Neuron classification: The authors should perhaps provide additional details on how the neuron types (FBI, FFI, pyramidal) were identified. I understand from the paper the dataset taken from that opto-tagging nicely identified the interneurons (and those could be separated based on laminar location). However, the waveforms of interneurons and pyramidal cells in the original publication look quite similar (Figure 3c of Bolding and Franks, 2018). Is it possible that interneurons could have been wrongly classified as pyramidal cells, perhaps in cases where units were further away from the optical fiber and might have not received enough light to be reliably driven? An additional analysis of waveform characteristics could increase confidence in the classification. Further, it is not entirely clear from the current manuscript whether the cell type classification was taken over from Bolding and Franks, or whether the authors ran their own classification.

2. Figure 1f: The authors should mention how many neurons of the different cell types were included in the analysis. In the original Bolding and Franks paper only 13 FFI were identified, but the figure legend states that 15 sessions were analyzed. This should be clarified.

3. Figure S3c: FFI could be mentioned in the legend as they are shown in the figure panel.

4. Figure 3A: Could the authors comment on why the induced γ response seems weaker in TeLC contra versus Thy-control? The difference is not significant (Figure 3C), but this might be an effect of the relatively low number of experiments in each group. In a purely feedback γ generation mechanism one might naively expect power to be equal in both controls.

5. Figure 5C: Principal cells seem to couple to distinct respiration phases (as mentioned also in the text). It would be informative to quantify whether individual cells maintain a similar preferred phase over time (as implied by the plot in Figure 5C), perhaps by taking tuning functions made from odd vs even breath cycles and cross-correlating them.

6. The authors argue that '[…] assembly recruitment would depend on excitatory-excitatory interactions among winner cells […]' (line 399). An alternative scenario that does not necessarily involve recurrent connections within the piriform cortex could be that OB drive activates a subset of principal cells that are most excitable (winner), which silence the less excitable population (losers) by recruiting FBIs (Pubmed ID 19515917). The authors could consider discussing along these lines in addition to what is already mentioned in the discussion.

---

## [Author Response]

Essential revisions:All points raised by the reviewers are for clarification and should be addressable either with textual adjustments or minimal additional analysis.While I would ask you to address all comments with textual clarifications, please do in particular consider the first two points made by reviewer 1 (directly γ-cycle dependent decoding – if feasible; decoding in the TeLC recordings, as similarly raised by reviewer 2) and the first and fifth comment of reviewer 2 (different populations and the TeLC comment as for Reviewer 1).

We thank the editors for their work and the reviewers for the constructive comments. As detailed in our answers below, we have performed several additional analyses to address the points raised. These led to the elaboration of new figure panels and associated text. In particular, we have performed a direct analysis of odor decoding within γ cycles as well as further analysis of the TeLC recordings. We have also strengthened the connection between feedback interneurons and γ oscillations by investigating the spike-triggered γ averages and performing current-source density analysis. Below we provide point-by-point answers to all points raised.

Reviewer #1 (Recommendations for the authors):– Does the observation that winner cells fire before the peak of the γ oscillation imply that γ is unimportant for neural coding and it is the deviation from the γ cycle that matters most?

The point raised by the reviewer is important and provocative, as our results suggest that the neurons that better encode a given odor can “deviate” or “escape” from the ongoing γ oscillation. Crucially, we show that winner cells are likely to recruit this inhibition and thus trigger the γ oscillation. We thus agree that this provocative idea is plausible and included new text in the Discussion hypothesizing that the neurons that fire tightly locked to γ are likely not involved in representing the odor sampled (page 10).

– The authors should be more explicit in the discussion of the initial results (Figures 1 and 2) that they are analyzing activity only during "spontaneous activity", i.e., recorded in the absence of odor. It took me a while to figure this out.

We apologize for not being clear enough in the description of both figures. Actually, the original versions of Figures 1 and 2 were computed using the entire recordings during the awake state, which includes both the odor stimulations and the spontaneous activity. In the revised version, we have recomputed Figures 1 and 2 panels only using spontaneous activity, which rendered virtually identical γ results (as expected since spontaneous cycles greatly outnumber odor cycles), and also edited the text to explicitly inform about this approach (page 4). Note, however, that by only analyzing odorless activity, β oscillations disappeared from the spectrum, reflecting their high odor dependency. Nevertheless, we have also modified Figure 4 to better discuss the dependence of γ and β oscillations on odor vs. odorless sniffing cycles.

– Light-evoked γ in contralateral hemispheres is much lower than in contralateral control hemispheres but never discuss this. It warrants mention.

We thank the reviewer for pointing this out. In the revised version, we now mention in the manuscript the different levels of light-evoked γ responses between TeLC contralateral and Thy control data that are apparent upon visual inspection of Figure 3 results. We now discuss possible network effects for such (potential) differences in γ responses, while also not discarding possible species-specific influences (given that the animals have different genetic backgrounds; please see page 6). Of note, we have also complemented panel A of Figure 3 by including a schematic representation of the experiments.

– Line 125: Confusing, FFIs in L. 1 are deeper but still more superficial than FBIs.

We thank the reviewer for spotting the mistake, which has been corrected. The sentence now reads (page 5):

“Additionally, VGAT+ neurons were further classified into feedback inhibitory interneurons (FBI) and feedforward interneurons (FFI) according to their location relative to the principal cell layer (FFI are located at layer 1 while FBI tend to be located within the cell layer 2/3; Bolding and Franks, 2018; Figure 1—figure supplement 5).”

We have also added a new supplementary figure showing the localization of each neuron subtype as well as their average waveform (new Figure 1—figure supplement 5).

– There is a small but noticeable β peak coinciding w/ peak of FFI firing (Figure 1F). This is worth mentioning.

We thank the reviewer for calling our attention to this observation; however, now that we computed Figure 1 using only the spontaneous activity, the small β peak disappeared. Nevertheless, we now explicitly mention in the discussion that the β peak timing matches that of FFIs (page 9).

– Does it matter that the authors selectively examined activity in VGAT- neurons, as opposed to the unsorted population? Would their observations hold (e.g. winner vs. loser cells) if they did not have this sorting?

To address this point, we performed the assembly analysis considering all neurons (i.e., without separating principal cells from inhibitory interneurons). We found reasonably similar results in the distribution of assembly weights, in the winners/losers activity time course, and in the correlation of assembly weights between odors (please compare Author response image 1 to the corresponding ones in Figure 6).

**Author response image 1. sa2fig1:** 

Nevertheless, we note that the much higher number of principal cells compared to interneurons is likely to account for this similarity (858 VGAT- vs 56 VGAT+). Such observation is now mentioned in the revised manuscript (page 15).

Reviewer #2 (Recommendations for the authors):1. Neuron classification: The authors should perhaps provide additional details on how the neuron types (FBI, FFI, pyramidal) were identified. I understand from the paper the dataset taken from that opto-tagging nicely identified the interneurons (and those could be separated based on laminar location). However, the waveforms of interneurons and pyramidal cells in the original publication look quite similar (Figure 3c of Bolding and Franks, 2018). Is it possible that interneurons could have been wrongly classified as pyramidal cells, perhaps in cases where units were further away from the optical fiber and might have not received enough light to be reliably driven? An additional analysis of waveform characteristics could increase confidence in the classification. Further, it is not entirely clear from the current manuscript whether the cell type classification was taken over from Bolding and Franks, or whether the authors ran their own classification.

We thank the reviewer for the suggestion. We apologize for not being explicit enough regarding the classification of neurons. We had to perform the classification anew based on the methodology described by Bolding and Franks since the labels employed in the original publication were unfortunately not available in the dataset. We explicitly inform in the revised version that we have run a new classification (page 14). We are now also including a supplementary figure (Figure 1-Supplement Figure 5) showing the average waveform of each neuron subtype and their position distribution in the probe. The waveforms and positions we obtained were similar to those presented in the original publication, though the exact neuronal numbers slightly differed: the original publication reported 855 VGAT-, 46 FBI, and 13 FFI. We obtained 858 VGAT-, 40 FBI, and 16 FFI (though 3 FFI were discarded due to an unusually low firing rate for an interneuron [<1 Hz]). We believe this to be a good match, given that we performed an independent classification. Importantly, we also obtain similar results if we compare the spiking frequency for each neuronal subtype, as shown in Author response image 2 (ours on the left, Bolding and Franks on the right).

Of note, in addition to the firing rates, the waveforms obtained by us (Figure 1-Supplement Figure 5) also match those obtained by Bolding and Franks (their Figure 3C).We agree with the reviewer that the waveforms are not as dissimilar as those obtained by classification methods based on the extracellular AP wave shapes. Nevertheless, to the best of our knowledge, the opto-tagging approach would be a superior method for neuronal classification. Moreover, we further note that all opto-tagged interneurons showed higher firing rates than the principal cells, thus further confirming their putative identity. Finally, we also note that Bolding and Franks showed in their Figure S2 that there are detectable differences among the waveforms.

2. Figure 1f: The authors should mention how many neurons of the different cell types were included in the analysis. In the original Bolding and Franks paper only 13 FFI were identified, but the figure legend states that 15 sessions were analyzed. This should be clarified.

We thank the reviewer for pointing that out. However, we note that – to obtain the result shown in the previous Figure 1F panel – we first computed the normalized spike rate for each session individually (and did not pool all neurons of a given subtype), that is, each sample was a session and not a neuron. In any case, in the revised version we now use each neuron as a sample and inform in the figure caption the number of analyzed neurons for each subtype.

3. Figure S3c: FFI could be mentioned in the legend as they are shown in the figure panel.

We thank the reviewer for spotting this omission, which has been fixed it. We note that this figure is now Figure 1—figure supplement 4.

4. Figure 3A: Could the authors comment on why the induced γ response seems weaker in TeLC contra versus Thy-control? The difference is not significant (Figure 3C), but this might be an effect of the relatively low number of experiments in each group. In a purely feedback γ generation mechanism one might naively expect power to be equal in both controls.

In the revised version, we now comment on the different levels of γ responses between TeLC contralateral and Thy control data that is apparent upon visual inspection of Figure 3 results (page 6). However, as noted by the reviewer, which we agree, the sample sizes for these experiments are relatively low to allow for conclusive inferences (5 for controls and 7 for TeLC contra). In any event, we now discuss possible network effects for such (potential) differences in γ responses (page 6), while also not discarding possible species-specific influences (given that the animals have different genetic backgrounds). We have also complemented panel A of Figure 3 by including a schematic representation of the experiments to help the reader understand the experimental conditions

5. Figure 5C: Principal cells seem to couple to distinct respiration phases (as mentioned also in the text). It would be informative to quantify whether individual cells maintain a similar preferred phase over time (as implied by the plot in Figure 5C), perhaps by taking tuning functions made from odd vs even breath cycles and cross-correlating them.

We thank the reviewer for the suggestion. We now computed the preferred resp phase for all principal neurons during the first and last third of the recording session. We find that most neurons maintain their phase preference. This result is now part of panel C of revised Figure 5 and mentioned on page 7.

6. The authors argue that '[…] assembly recruitment would depend on excitatory-excitatory interactions among winner cells […]' (line 399). An alternative scenario that does not necessarily involve recurrent connections within the piriform cortex could be that OB drive activates a subset of principal cells that are most excitable (winner), which silence the less excitable population (losers) by recruiting FBIs (Pubmed ID 19515917). The authors could consider discussing along these lines in addition to what is already mentioned in the discussion.

We thank the reviewer for the thoughtful comment and excellent suggestion. After analyzing odor assemblies in TeLC recordings (please see our answers above), we found that the alternative scenario mentioned by the reviewer turns out to be more likely than our previous hypothesis. This is because we found that the TeLC infection does not alter winning neurons; instead, the TeLC lesion significantly decreases the number of losers and attenuates their inhibition during the γ window (please see our answers above). Therefore, we changed this sentence to match our new results. It now reads (page 9):

“(…) the assembly recruitment would depend on OB projections determining which winner cells “escape” γ inhibition, highlighting the relevance of the OB-PCx interplay for olfaction (Chae et al., 2022; Otazu et al., 2015).”